# Essential Trace Elements in the Shells of Commercial Mollusk Species from the Black Sea and Their Biotechnological Potential

**DOI:** 10.3390/ani15111637

**Published:** 2025-06-02

**Authors:** Larisa L. Kapranova, Juliya D. Dikareva, Sergey V. Kapranov, Daria S. Balycheva, Vitaliy I. Ryabushko

**Affiliations:** A.O. Kovalevsky Institute of Biology of the Southern Seas of RAS, 2 Nakhimov Ave., Sevastopol 299011; lar_sa1980@mail.ru (L.L.K.); dikareva.julija@rambler.ru (J.D.D.); sergey.v.kapranov@yandex.ru (S.V.K.); rabushko2006@yandex.ru (V.I.R.)

**Keywords:** essential trace elements, shells, bivalves, gastropods, *Anadara kagoshimensis*, *Crassostrea gigas*, *Mytilus galloprovincialis*, *Flexopecten glaber*, *Rapana venosa*, Black Sea

## Abstract

Mollusk shells, previously regarded as waste, are currently attracting attention as valuable resources in various applications, including food additives production. The present research focuses on the potential use of shells of five commercially important Black Sea mollusks as a source of essential (physiologically indispensable) trace elements in human and animal nutrition. The shells of different mollusks sampled from the same site exhibited differing levels of enrichment in specific elements. The highest concentrations of chromium, manganese and iodine were found in shells of the clam *Anadara kagoshimensis*; oyster *Crassostrea gigas* shells exhibited the maximum enrichment in iron and cobalt; copper and selenium were most concentrated in gastropod *Rapana venosa* shells; the highest concentration of zinc was observed in mussel *Mytilus galloprovincialis* shells; and the highest molybdenum concentration was noted in scallop *Flexopecten glaber ponticus* shells. Species specificity was also detected in the patterns of pairwise positive and negative correlations of elements in each mollusk. A few tens of grams of ground shells of *Anadara*, *Crassostrea*, and *Rapana* were found sufficient to cover the daily human requirements for many essential trace elements.

## 1. Introduction

Mollusk aquaculture (malacoculture) is an industry with great potential for the production of both food and non-food products [1,2]. When processing mollusks, the shells account for approximately 60–80% of the total mass, while the edible parts comprise only 12–17% [3]. Many trace elements are concentrated in the shells, with the highest shares for Ni and Mn exceeding 80% and the shares for Fe, Zn, Cu, Se, Co being 63–94% [4]. The bioaccumulation of trace elements, i.e., elements with concentrations of below 100 ppm [5], in the shells of mollusks is related to their habitat and occurs largely due to the processes of biomineralization and adsorption [3,6]. Moreover, the concentration of elements in mollusk shells depends on abiotic environmental factors and may differ in different biotopes [7].

Information on essential trace elements (ETEs) in mollusks, including their shells, is incomplete, not least due to the ambiguous definitions of element essentiality [8,9,10,11]. According to the classification developed by the World Health Organization [12], trace elements exhibiting positive physiological activity are divided into three groups: (1) essential, (2) probably essential, and (3) potentially toxic with possibly essential functions. The first group includes Fe, Co, Cr, Cu, Zn, Se, Mo, and I. The second one comprises B, Si, V, Mn, and Ni. The third group consists of the elements Li, F, Al, As, Cd, Sn, and Pb.

After the soft tissues have been taken out, the shells of mollusks often remain in the environment, where they decompose very slowly and alter the ecosystems. Without regulated disposal procedures, shell waste is frequently discarded in landfills or dumped at sea. It accumulates and pollutes coastal areas, emitting an unpleasant odor due to the decomposition of the organic matter trapped within the shells [1,13,14]. The effects of increased shell deposition beneath collectors and shell accumulation on shellfish farms are not well understood, but these environmental impacts can be both positive and negative [15]. On the one hand, the accumulated shells provide a habitat for macrozoobenthos. Mollusk shells serve as a substrate for the attachment of epibionts and shelter from predators, and they affect the transport of materials in the benthic environment [16]. On the other hand, fragmented shells reduce animal diversity and abundance, e.g., by preventing oxygen from penetrating into the underlying sediments. The impact of suspended oyster farming and the accumulation of shells on the seafloor on the burrowing ability of the polychaete *Perinereis aibuhitensis* was studied in [17], which showed the inhibition of the burrowing activity of *P. aibuhitensis* by unprocessed oyster shells.

Therefore, there is a need for various alternative ways of utilizing mollusk shells, including using them as a source of essential elements [1,14,18]. In addition to being usable as biofilters, adsorbents, deacidifying agents, and building materials, mollusk shells find applications as food/feed additives and nutraceuticals [19,20,21,22]. It is important to take into account that marine organisms are capable of bioconcentrating heavy metals along the food chain [23]. Therefore, consuming seafood products containing heavy metal concentrations that exceed permissible levels may pose a health risk. This applies to essential elements as well: even these elements can be toxic if taken in excess amounts [24,25,26]. To minimize the risks associated with shellfish consumption, it is advisable to monitor the levels of metals in shellfish products. This monitoring allows for comparison with established safe limits for the concentration and dosage of elements in the food consumed. For example, the safe concentration for Cd is (1.01–2.00 μg/g), while a concentration of 5.01–10.00 μg/g is considered hazardous [27]. The minimum safety limit for zinc is 100.01–250.00 μg/g, and the highest safety limit is 667.01–750.00 μg/g [27].

Formulations containing mollusk shell powder have long been used in traditional Eastern medicine to treat skin lesions, inflammations, rheumatism, and stomach ulcers [22]. Bioactive products that can be produced from mollusk shells include antibacterials, antioxidants, and antihypertensive and anti-hypercholesterolemic agents, as well as calcium-enriched supplements to enhance bone health, improve blood circulation in livestock, and increase the quality of egg shells in egg-laying poultry [20]. The use of shellfish by-products is expected also to contribute to a more sustainable approach to supplying essential micronutrients in feed additives or encapsulated powdered functional products for humans. Therefore, it is important to know the concentration of essential trace elements in mollusk shells in order to calculate the dosages required to fulfill daily human requirements.

Data on the ETE concentrations in the biota of the Black Sea are scarce and unsystematic. For instance, the trace element profile of the tissues and shells of the Black Sea scallop *Flexopecten glaber ponticus* is virtually unstudied, with all available publications on the elemental composition of this species being associated only with the Mediterranean subspecies from the Gulf of Taranto and the Dardanelles Strait [28,29,30]. In contrast, trace element concentrations in tissues of the mussel *Mytilus galloprovincialis* have been studied quite extensively (see [2,31,32,33,34] and references therein), and over the past decade, more than 20 articles have been published on trace elements in mussels from the Black Sea region [34]. There exists some information on trace element concentrations in the meat of the veined rapa whelk *Rapana venosa* and the oyster *Crassostrea gigas* (see [33,35] and references therein). However, none of the available publications reported the concentrations of the full suite of ETEs in Black Sea mollusk shells.

The goals of this work are to determine the concentrations of the essential trace elements Cr, Mn, Fe, Co, Cu, Zn, Se, Mo, and I in the shells of five species of mollusk from the Black Sea: the bivalves *Anadara kagoshimensis* (Tokunaga, 1906), *Crassostrea gigas* (=*Magallana gigas*) (Thunberg, 1793), *Flexopecten glaber ponticus* (Bucquoy, Dautzenberg & Dollfus, 1889), and *Mytilus galloprovincialis* (Lamarck, 1819) and the gastropod *Rapana venosa* (Valenciennes, 1846); to assess species specificity in ETE accumulation in their shells; and to calculate the amounts of powdered shells needed to satisfy daily human requirements for ETEs.

## 2. Materials and Methods

### 2.1. Sampling Site and Sampled Mollusks

To exclude the influence of local non-uniformities in the element distribution in the environment, mollusks were collected from one biotope—at a shellfish farm at the exits from Karantinnaya and Sevastopol Bay (Black Sea) in the coastal area of the city of Sevastopol (44°36′56.4″ N; 33°30′13.6″ E) (Figure 1). The farm water area does not experience significant anthropogenic pressure throughout the year, as there are no industrial enterprises in the vicinity of the farm. The source of pollution may be currents from Sevastopol Bay that carry pollutants outside it [36]. In the marine farm area, sedimentation processes make a substantial contribution to the water purification by removing a substantial quantity of toxic elements [37].

All mollusks were collected in October 2022 at a surface water temperature of 21.4 °C and a salinity of 18.39. These values were somewhat higher than the mean ones recorded in October over the period of 2001–2018 (temperature 17.5 °C and salinity 17.80), and they were even above the maximum limits for that period (temperature 18 °C and salinity 17.96) [36]. The high temperature and salinity values are due to the steadily increasing trends of these parameters over the past few decades in this region [36]. The bivalves were collected from plastic cages at a depth of 2–3 m. The age of the mollusks was two years from the time their larvae settled in the cages. The gastropod individuals were collected beneath the farm collectors at a depth of 18 m (at a temperature of 19 °C) by scuba diving. For the study, shells of 6–10 individuals of each species were used. The shell lengths of the selected bivalve individuals were as follows: 28–32 (30.5 ± 1.0) mm for *Anadara kagoshimensis*, 93–100 (90.7 ± 10.3) mm for *Crassostrea gigas*, 31.1–45.5 (38.8 ± 5.1) mm for *Flexopecten glaber ponticus*, and 50.0–54.5 (54.1 ± 2.9) mm for *Mytilus galloprovincialis*. The shell height of *Rapana venosa* was 77.5–89.4 (84.7 ± 4.8) mm. All the values are expressed as min–max (mean ± SD). The sex and the gonadal ripening stage in the mollusks were not determined.

### 2.2. Analytical Sample Preparation

The mollusks collected from their habitat were delivered to the laboratory in plastic buckets of seawater within 1–2 h after the sampling and processed immediately after the delivery. The shells were cleaned from epifauna and epiphytes with a ceramic scraper and a plastic brush. The shells of the bivalve mollusks were opened using a scalpel, and the shells of the gastropods were broken mechanically. After the thorough separation of soft tissues from shells with a plastic scalpel, the shells were rinsed with deionized water and dried in an oven at 105 °C for 5 h.

The dried shells of each individual were crushed separately in a porcelain mortar, and 100 mg of the shell powder was transferred to the acid digestion PTFE tubes that had been preliminarily soaked in 10% nitric acid, rinsed with deionized water, and dried. Nitric acid of analytical grade was subjected to sub-boiling distillation in an acid purification system DST-1000 (Savillex, Eden Prairie, MN, USA), and 2.00 mL aliquots of the distilled nitric acid were added into the tubes, which were closed with PTFE screw caps and left to stand overnight. The next day, the digestion tubes were autoclaved at 120 °C for 2 h, and the digested samples were diluted with deionized water in polypropylene tubes to a dilution factor of 1000 mL g^–1^ dry weight (d.w.).

### 2.3. ICP-MS Analysis

Elemental analyses were performed using inductively coupled plasma mass spectrometry (ICP-MS) on a single-quadrupole instrument PlasmaQuant MS Elite (Analytik Jena, Jena, Germany). The instrument was preliminarily calibrated using the Analytik Jena tuning solution for ICP-MS and the internal Resolution and Trim (with 1 ppb Be and Th) and Mass Calibration (with 1 ppb Be, Mg, Co, In, Ce, Ba, Tl, Pb, and Th) procedures. The instrument settings were the same as in [33]. For each analytical sample, there were 5 scans per replicate and 5 replicates per sample. To obtain the analytical sample mean, these values were averaged.

The calibration curves were obtained using standard solutions prepared by the dilution of a certified multielement standard, IV-ICPMS-71A (Inorganic Ventures, Christiansburg, VA, USA), and a certified iodide standard, GSO7620-99 (Lenreaktiv, Saint Petersburg, Russia), with deionized water to the analyte concentrations of 0.1, 1, 5, 10, 50, 100, and 1000 μg L^–1^. The coefficients of determination of the resulting linear calibration fits were no smaller than 0.999. The measured concentrations of the elements, except Fe, in all the samples fell within the limits of the calibration curves. For measuring iron concentrations, the samples were additionally diluted by a factor of 10 to 20. The limits of detection were below 0.3 μg L^–1^ (Se). No internal standard was used in the measurements. To correct for the signal drift over time, the apparent concentrations in the 10 μg L^–1^ standard solution were measured after every fifth sample, and a time-dependent piecewise linear function of the standard solution signal was used for the concentration correction.

To eliminate the significant calcium matrix effects, the optimal dilution procedure was implemented. An acid-digested shell sample was progressively diluted with consecutive increases of the dilution factor from 250 to 1500 mL g^–1^ and a step of 250 mL g^–1^. The dilution factor that resulted in no further decrease in the apparent concentrations of all the analytes (on a dry weight basis) was adopted as the optimal one. In the trials, it was estimated at ~1000 mL g^–1^.

### 2.4. Quality Assurance/Quality Control

The quality assurance and quality control procedures applied were quantifying trace elements in a certified reference material, duplicate sampling, and the standard additions.

The entire sample preparation procedure and the ICP-MS analyses were repeated for samples of European Reference Material, ERM^®^-CE278k (*n* = 5). The element concentration recoveries were 87–116% compared to the certified values.

To increase the measurements’ accuracy and reliability, each individual shell sample was prepared and analyzed in duplicate, and the arithmetic mean was adopted as the expectation value.

The validation using standard additions was run to ensure the validity of the dilution procedure and the elimination of matrix effects. To this end, the concentrations of all the analytes, except iron, in a diluted sample were increased by 0.20, 0.40, 1.00, 5.0, 10.0, 50, and 100 μg L^–1^ through additions of appropriate amounts of diluted certified standards. For iron, the added concentrations were 200, 400, 600, 800, and 1000 μg L^–1^. The total measured concentrations comparable with the initial one (0 μg L^–1^ added) were subjected to linear regression in GraphPad Prism 8.0.1. The initial concentrations were found to fall within the 95% confidence intervals for the corresponding X-intercepts, which indicated the validity of the dilution procedure and the proper elimination of matrix effects.

### 2.5. Statistical Analysis

The statistical comparison of the element concentrations in the shells of different mollusks was performed in PAST 4.14. The lack of dispersion heterogeneities was tested using Levene’s test. In case of a lack of dispersion heterogeneities, the significance of differences between the samples was tested using classical Fisher’s ANOVA supplemented with the Tukey’s pairwise test; otherwise, Welch’s ANOVA and the Games–Howell pairwise test were applied. In multiple comparisons, the Benjamini–Hochberg correction was applied to the unadjusted *p*-values [38]. The significance limit was adopted at *p* = 0.05.

A principal component analysis (PCA) and a hierarchical cluster analysis were run in PRIMER 6.1.16 and PERMANOVA+ 1.0.6 with the Euclidean distance as a similarity measure. To analyze the results in equal metrics, they were subjected to Z-normalization (or Z-standardization):
(1)
$$Z_{i}=\left(x_{i}-\bar{x}\right)/SD\left(x\right)$$

where 
$\bar{x}$
 is the mean value of the quantity, 
x
, and 
$SD\left(x\right)$
 is its standard deviation.

In the cluster analysis, the unweighted pair group method with the arithmetic mean (UPGMA) was used as clustering algorithm, and a SIMPROF test was used to check the separation of clusters at the 5% significance level. In the dendrograms, black solid lines denote significant separation, and red dotted lines denote the lack of significant separation of clusters.

Spearman’s correlation coefficients (*r*_s_) were calculated in PAST 4.14, with a two-tailed *t* test with *n* − 2 degrees of freedom used to estimate the probability of non-zero *r*_s_:
(2)
$$t=r_{s}\sqrt{\frac{n-2}{{1-r}_{s}^{2}}}$$


## 3. Results

### 3.1. Element Concentrations and Multivariate Analyses

The results of the analysis of essential trace elements in Black Sea mollusk shells are presented in Table 1. 

The elements formed the following descending order of concentrations, averaged over all the species: Fe > Cu > I > Mn > Zn > Cr > Se > Co > Mo. The highest concentrations of Cr, Mn, and I were observed in *A. kagoshimensis* shells; Fe, Co, and Cu were most abundant in shells of *C. gigas*; Zn and Se were most concentrated in shells of *M. galloprovincialis* and *R. venosa*, respectively; and *F. glaber ponticus* shells shared nearly equally high mean concentrations of Mo with those of *A. kagoshimensis*. Among the elements, the iron in oyster shells exhibited the highest concentration, which exceeded 1.3%. The least abundant was molybdenum in mussel shells (0.3 ppm).

The hierarchical cluster analysis of the log-transformed concentrations for all the species (Figure 2a) showed that the elements form three distinct clusters, separated according to the overall element levels. The uppermost level contains a singleton cluster representing iron (10^3^–10^4^ mg kg^–1^). The second major cluster encompasses the Cu, Zn, Mn, and I leaves (10^1–^10^2^ mg kg^–1^), and these individual element clusters are significantly separated from one another. The third major cluster represents Cr, Se, Co, and Mo (10^–1^–10^1^ mg kg^–1^), and the individual element clusters are not significantly separated, according to the SIMPROF test results.

The cluster analysis applied to the Z-standardized, log-transformed concentrations (Figure 2b) revealed a markedly different case. The resulting dendrogram consists of two upper-level clusters, with one of them representing chromium. Another one is split into a selenium leaf and a ramified cluster representing the rest of the elements. A similar distinctive feature is also clearly seen in the principal component analysis biplot (Figure 3), in which projections of the Cr and Se vectors on Principal Component 1 (PC1) have negative values, while projections of all the other vectors are positive. Chromium and selenium have specific distinctive features in the analyses of Z-standardized values, as the lowest concentrations of these elements are associated with *C. gigas* shells, which contain high concentrations of the other elements. It is thus evident that oysters are very effective at accumulating most of the metallic elements and iodine in their shells (or at preventing Cr and Se accumulation in them).

Principal Component 1, which explains 34.5% of the total variance, is mainly related to the overall element accumulation in shells. In contrast, Principal Component 2 (PC2), which contributes 21.9% of the total dispersion, is largely associated with the species-specific element accumulation, as there is a considerable degree of data separation in the positive direction of PC2 from *A. kagoshimensis* to *C. gigas*.

The Fe, Cu, and Co vectors in Figure 3 expectedly point toward the *C. gigas* data as the oyster shells have the highest concentrations of the corresponding metals (Table 1). Similarly, the vectors of Cr, Mn, Mo, and I are directed mainly to the data of *A. kagoshimensis*, indicating the enrichment of ark clam shells with these elements. The Se and Zn vectors have a pronounced negative PC3 component, and it is difficult to identify the specific groups of data associated with these vectors.

### 3.2. Correlations

Graphical matrices of Spearman’s correlation coefficients for the element concentrations in the mollusk shells are shown in Figure 4. Most of the positive correlations are noted for *F. glaber ponticus*, with the significant ones being in pairs: Co–Zn, Mo, I and Zn–Mo, I. Most of the negative correlations are recorded for *M. galloprovincialis*, especially for Se and Zn, but no correlations are significant for this mollusk. There are two significant positive correlations for *A. kagoshimensis* (Cr–Mo and Co–Cu) and for *C. gigas* (Zn–Fe and Zn–Se), and three positive correlations are significant for *R. venosa* (Mn–Co, Zn–Cu, and Zn–I).

It can be seen that the correlation matrices bear little resemblance to one another. This observation enables the differentiation of one mollusk species from another through the use of the correlation coefficients and multivariate techniques, e.g., cluster analysis. The cluster analysis of Spearman’s correlation coefficients arranged in five columns corresponding to each mollusk showed the significant differences across all the species based on the SIMPROF test results (Figure 5). Most of the similarities are between the shells of *A. kagoshimensis* and *R. venosa*. The most differing species in these terms is *C. gigas*.

## 4. Discussion

### 4.1. Factors Affecting the Element Accumulation in Mollusk Shells

Mollusks, like other marine organisms, have the ability to accumulate trace elements from the aquatic medium [39,40,41]. The process of element uptake is significantly affected by the speciation of these elements and their presence in either dissolved or suspended states in the environment. Notably, the concentrations of trace elements in mollusk soft tissues are many orders of magnitude greater than those in the aquatic environment [42].

The trace element content in mollusk shells is commonly believed to be determined by the environmental conditions, including surrounding trace element levels [43,44,45,46]. However, biological factors (e.g., metabolic activity and respiration rate) have been shown to affect the element accumulation as well [43,47,48], but the underlying mechanisms are poorly understood to date. The species specificity of trace element deposition in shells of different mollusks from the same habitat has been noted in several studies [49,50], also suggesting the presence of biological control.

There is evidence that the incorporation of different trace elements into shell is governed by a common process [51], and it has been suggested that the processes controlling the concentration of elements in the extrapallial fluid mediating shell formation are dominant over element-specific mechanisms of element embedding in the shell matrix. At the same time, it was shown that some metals, e.g., Cd, can bind to the shell surface by precipitation and chemisorption, without any shell formation process involved [52].

Soft tissues of marine invertebrates, including mollusks, are a good source of essential trace elements in human nutrition [2,33,53,54,55], but there are additional aspects that should be taken into account when considering each particular species as a food item. One of them is the concomitant intake of toxic elements, whose concentrations or daily/weekly doses should be estimated and compared with the maximum permissible levels to prevent health risks [56]. Another one is the bioaccessibility of both essential and toxic elements in the food [57,58,59]. At present, no such information regarding trace elements in mollusk shells is available.

### 4.2. The Comparison of the Results with the Literature Data

Comparing the concentrations of elements in the shells of *Anadara kagoshimensis* from the Black and Azov Seas, it can be concluded that the mollusk from the Black Sea accumulates more Cr, Mn, Co, Cu, Zn, Se, and Mo in its shell, while the same mollusk from the Azov Sea accumulates predominantly I and Fe [60], likely reflecting the effect of different salinities. Since there is no information in the literature on the elemental composition of *A. kagoshimensis* shells from other regions, a closely related species, *A. granosa*, can be considered. In a study by Bharatham et al. [61], element concentrations were analyzed in *A. granosa* shells collected in Malaysia. The concentrations of Fe (595.46, here and below, all values are in mg·kg^–1^) and I (10.14) were lower than in our study (1564 and 55.6). However, the concentrations of Mn (255.38) and Cu (93.20) were significantly higher than those found in the present research (35.1 and 75.0, respectively). *A. granosa* shells from Kuala Juru, Kuala Kurau, and Jeram, Malaysia [62], contained significantly less Fe (311.98) and Cu (4.95) compared to our study results. The Zn values (7.67) did not differ significantly from the results of the present work (8.85).

The elemental concentrations in shells of the Black Sea scallop *Flexopecten glaber ponticus* have not been studied previously. In this regard, in our study, we turned to a related species, *Aequipecten opercularis*, collected from the Sacca di Goro area in the northern Adriatic Sea, Italy [63]. A comparison of the obtained data with the results of our analysis showed that the concentrations of Fe (1592.27), Co (0.496), and Cu (6.8) in the shells of the scallop from Italy were lower than those in the Black Sea species (2285, 1.29, and 89.4). At the same time, the levels of Mn (75.53) and Zn (41) in the samples from the Adriatic were significantly higher.

In a study by Cardoso da Silva et al. [64], it was found that shells of Brazilian oysters *Crassostrea brasiliana* and *C. mangle* accumulate Fe (0.0029), Co (0.082), and Zn (4) in lower concentrations compared to the species from the Black Sea (13157, 2.73, and 21.1, respectively). However, the concentrations of Cr and Mn did not exhibit significant differences between samples from the Brazilian coast (0.72 and 28) and from the Black Sea (0.22 and 24.3).

Sereanu et al. [65] investigated the effect of habitat on elements in *Rapana venosa* shells from three sites on the Romanian shelf. The comparison with the present study shows that the *Rapana* shells from the Romanian coast accumulated significantly lower concentrations of Mn (2.96), Fe (9.34), and Zn (1.03) compared to those from the Crimean coast (16.2; 3294; and 6.81, respectively). However, the concentration of accumulated Mo (1.26) in the *Rapana* shells from the Romanian shelf exceeded the Mo concentration in the Crimean ones (0.56). In work [66], the Cu and Zn concentrations were analyzed in shells of *Mytilus galloprovincialis* and *R. thomasiana* from the Romanian coast of the Black Sea. The Zn concentration in shells of *M. galloprovincialis* and *R. thomasiana* were 22.99 and 16.82, and the Cu concentrations were 10 and 5.88, respectively. In the present study, the zinc concentrations in shells of *M. galloprovincialis* and *R. venosa* were 30.0 and 6.81, while the copper concentration in mussel shells was more than twice as low as that in *R. venosa* shells (77.1 and 190, respectively).

Concentrations of Mn (1.9), Fe (155), Co (0.1), and Zn (4) in shells of *M. galloprovincialis* from South Africa [7] were lower than those in mussel shells from Black Sea (8.17, 1962, 0.99, and 30.0, respectively), while the I and Cr concentrations were comparable (5–13 and 1.4–1.6 vs. 5.90 and 1.69). The Mn concentration (54.4) in shells of the Baltic Sea mussel *M. trossulus* [50] was found to be significantly higher than the value from the present study. The results of the element quantitation in the mussel shells sampled in the Black Sea in 2020 [67] showed that the Co concentration (35.6) was significantly higher than the corresponding value from the present work, while the Cu concentration (2.9) was much lower. The concentrations of Mn (8.2) and Zn (38.9) closely matched the results of this research.

### 4.3. Role and Requirements for Essential Trace Elements in Humans

Below, we characterize the role of essential trace elements in humans and provide the dosages of Black Sea mollusk shells that are sufficient to meet the daily human requirements for these elements, under an assumption of their 100% bioaccessibility, as rough estimates of the biotechnological potential of shells as essential trace element sources.

#### 4.3.1. Iron

Iron is vital to almost all living organisms because it is involved in a wide range of metabolic processes, including oxygen transport, DNA synthesis, and electron transport [68]. Iron is required as a structural component of hemoglobin, which is in the red blood cells that carry oxygen from the lungs to the tissues, and myoglobin, which carries and stores oxygen in the skeletal muscle tissues. This mineral is also essential for physical growth, brain development, cell functioning, and hormone production [69]. Up to 40% of the world’s population suffers from iron deficiency [70,71]. 

The highest concentration of Fe was found in the oyster shells; therefore, they can be regarded as a natural rich source of iron. Oyster shells are multi-layered structures made up of calcium carbonate crystals, along with protein and pigments [72,73]. The pigments of the shells of *C. gigas* are a complex composition of porphyrins and ommochromes [74,75], and it can be assumed that the oyster shell pigments form stable complexes with iron.

For adults, the average daily requirement is 7 mg Fe [76]. The results obtained in the present study show that 1 kg of oyster shells contains 13.2 g of iron, which suggests that consumption of 0.532 g of ground shells is sufficient to meet daily human requirements.

#### 4.3.2. Manganese

Manganese is a cofactor for a number of enzymes that regulate metabolism and cell energy, reproduction, and the growth of bones and connective tissue [77,78]. The daily intake of Mn is about 10 mg [76]. 

The elemental composition of mollusk shells depends largely on the presence of metal deposits in the periostracum [79]. In this case, Mn can perform a protective function against shell destruction when ambient acidity increases [80]. Apparently, the shells of *Anadara*, which in natural conditions burrow into the sediment, are predisposed to concentrating Mn in the organic layer of the shells.

The concentration of manganese in *Anadara* shells is significantly higher than in those of the oyster, scallop, mussel, and rapa whelk (Table 1). To cover the daily requirement for this element, it is necessary to consume 285 g of ark clam shells. The concentration of heavy metals in *Anadara* shells is very small [81], and they are safe for consumption as a micronutrient-enriched food additive; however, the large amount of shell powder needed for the fulfillment of daily Mn requirements renders mollusk shells an inappropriate source of this element.

#### 4.3.3. Zinc

Zinc is an essential micronutrient for invertebrates due to its role in enzymatic reactions and its protective effect against damage, such as that caused by heavy metals [82]. The distribution of Zn in mollusk shells depends on the geochemical conditions of the habitat, the physiological state of the organism and its food supply, as well as on the type of organic matrix of the shells [83]. Oysters are widely known for their high zinc concentrations [84] and are bioindicators of Cu and Zn levels in the environment [85,86,87]. In the present study, the highest concentration of Zn was found in mussel shells (Table 1). The concentration of Zn in oyster shells is lower than in mussel shells but an order of magnitude higher than in shells of *Anadara*, *Flexopecten*, and *Rapana*.

Zinc is involved in growth, development, reproduction, protein and carbohydrate metabolism, and in oxidation-reduction processes [88]. For adults, 15 mg of zinc is the required daily dose [89]. Zn deficiency can cause growth and developmental delays and cell apoptosis [90,91]. The highest concentration of zinc was found in mussel shells: 30.0 ± 15.4 mg kg^–1^. To meet the daily human requirement for zinc, approximately 500 g of mussel shells is the needed dose, which exceeds any reasonable limits, and mussel shells cannot serve as an adequate source of zinc for humans.

#### 4.3.4. Copper

Several billion people on Earth suffer from copper deficiency [92]. Cu is a cofactor for over 30 different enzymes; it is essential for blood circulation, is involved in energy metabolism, supports bone health, protects vision, promotes collagen production, and enhances immune defense [93]. Cu is a component of enzymes that have oxidation-reduction activity and participate in iron metabolism; it stimulates the assimilation of proteins and carbohydrates and is involved in the oxygenation of human tissues [94]. In addition, this trace element is a cofactor for lysyl oxidase and is necessary for the intermolecular bonding of collagen and elastin. Copper is the main component of the myelin sheath and is involved in collagen synthesis, skeleton mineralization, the formation of red blood cells, and the synthesis of skin pigments. The daily requirement for Cu in adult men and women is 1.55 mg [95].

The highest copper concentration was observed in the shells of the veined rapa whelk. This observation may be associated with the different shell structures of the mollusks. Since the shells of all mollusks contain an organic matrix that regulates biomineralization processes [96], it is believed that elements bound to the organic components are more easily assimilated by human or animal body.

To fulfill the daily Cu requirement, one should consume 8 g of ground *Rapana* shells, and thus, *Rapana* shells are a more suitable source of copper than, for example, mussel or ark clam shells.

#### 4.3.5. Cobalt

Cobalt is a component of vitamin B12, which is necessary for a number of enzymatic reactions and for the formation of red blood cells and nerve sheaths. It also strengthens the immune system and stimulates the activity of white blood cells to prevent infections [97]. Cobalt is involved in the secretion of the hormones insulin and adrenaline, stimulating the activity of the endocrine glands; it promotes the accumulation of a number of vitamins in the body and stimulates protein synthesis for muscle building. As a part of the cyanocobalamin molecule, Co participates in enzymatic processes, inhibits iodine metabolism, increases iron assimilation and hemoglobin synthesis, stimulates erythropoiesis, participates in the transport of oxygen in the body, and has the ability to restore the disulfide S-S groups involved in the binding and utilization of toxic compounds [98]. Only with the normal interaction of Co, Cu, and Fe does the process of hematopoiesis occur in animals and humans. Humans get vitamin B12 mainly from animal foods. Vitamin B12 is a stimulant of hematopoiesis; promotes the absorption of iron by the body; stimulates hemoglobin synthesis and the formation of protein complexes and thyroid hormones; and promotes the excretion of water by the kidneys.

The recommended daily intake for Co has not yet been established [98]. Adults consume an average of 5 to 50 μg Co per day [80]. A volunteer-based study showed that even a dosage of 1000 μg per day did not cause clinically significant changes in the health of adults [99]. A Co dose of greater than 30 mg/kg is considered lethal for animals [99].

Thus, to obtain the daily cobalt dose of 50 μg, it is necessary to consume, for example, 18.3 g of ground oyster shells. 

#### 4.3.6. Iodine

The average iodine consumption is 50–99 μg per day [100], which is 1.5–3 times less than the established requirement of 150 μg per day [101]. More than one billion people worldwide suffer from a deficiency of this micronutrient [102]. Iodine deficiency causes mental retardation, especially in children, but this consequence occurs only when the deficiency is severe.

The iodine-containing hormones thyroxine and triiodothyronine are involved in the regulation of the functional state of the central nervous system and the emotional state of a person, as well as the cardiovascular and musculoskeletal systems. They have a pronounced effect on water–electrolyte, protein, lipid, carbohydrate, and vitamin metabolism [103]; induce increased oxygen consumption in tissues; and regulate tissue differentiation, growth, and developmental processes, including neuropsychiatric ones.

The maximum concentration of iodine was found in the *Anadara* shells, and the minimum concentration was found in the mussels (Table 1). Thus, ark clam shells are a valuable source of iodine in human nutrition. In addition, the iodine in shells is likely chemically bound to organic matrix components, which enhance iodine bioaccessibility in the human body. To cover the daily requirement for this element, it is sufficient to consume 2.7 g of powdered ark clam shells.

#### 4.3.7. Chromium

Chromium is included in marine biogenic carbonates and participates in the biomineralization of shells [104]. In humans, chromium supports insulin production and blood sugar control; reduces the heart rate in people with impaired glucose tolerance; reduces inflammation; is involved in weight and free-testosterone regulation in patients with polycystic ovary syndrome; and normalizes appetite [105,106]. Furthermore, Cr^3+^ plays a role in the metabolism of lipids, carbohydrates, and proteins and influences the elimination of heavy metals and toxins from the body [107].

The recommended daily dose of Cr is 50–200 μg [108]. As the highest concentration of Cr was found in the *Anadara* shells, one should consume 25.6 g of ground ark clam shells to cover the daily human needs for Cr, which is quite large an amount. Therefore, it is reasonable to consider consuming chromium with food.

#### 4.3.8. Molybdenum

Molybdenum is an essential element for humans [109,110,111]. Molybdenum functions as a cofactor for enzymes in the metabolism of carbohydrates and lipids, which are involved in the generation of uric acid and facilitate its elimination [112,113]. It is also important for the growth of mollusks [114,115]. The sole known biochemical role of molybdenum in animal physiology, aside from its interaction with copper, pertains to the synthesis and activation of xanthine oxidase [116].

The concentration of Mo was slightly higher in the shells of the ark clam and the scallop (Table 1). The average required dose for humans is about 0.3 mg per day [76]. To cover the daily requirements for Mo, one would need to consume as much as 283 g (18 tablespoons) of ground shells. For this reason, mollusk shells cannot serve as a practical source of molybdenum for humans.

#### 4.3.9. Selenium

The functions of selenium in the human body are well known [117,118,119,120,121]. Selenium plays a crucial role in supporting the body’s immune, antioxidant, and detoxification systems, and it is active in decreasing the cancer risk. Additionally, this trace mineral is essential for regulating the thyroid and reproductive systems and helps lower the risk of heart disease [117].

Selenium helps protect shells from oxidative stress, having antioxidant properties [118]. In mussels, both Se and Zn play a role in influencing spawning, growth, and development [122]. Selenium serves to protect the body from reactive oxygen species and is crucial for maintaining spermatogenesis and sperm viability [123]. Although in *Rapana* shells the Se concentration appeared to be higher, there were no statistically significant differences from the concentrations in the shells of the other mollusks (Table 1).

The adequate Se intake level is 87 μg per day [124]. To cover the daily requirement for selenium, one needs to consume 23.1 g (1.5 tablespoons) of ground *Rapana* shells. The appropriate selenium intake in the form of powdered shell samples is beneficial; however, it should be kept in mind that excess selenium is toxic [125]. Therefore, it is important to accurately estimate the Se concentrations in each consignment of mollusk shells to be able to calculate the required dose of shell powder.

## 5. Conclusions

In the present study, the concentrations of essential trace elements (Cr, Mn, Fe, Co, Cu, Zn, Se, Mo, and I) have been determined in the shells of five common, commercially important Black Sea mollusks sampled from a shellfish farm. The highest concentrations of iron, nickel, and cobalt were found in the shells of the oyster *Crassostrea gigas*. The highest concentrations of zinc were observed in the shells of the mussel *Mytilus galloprovincialis*, and the highest concentrations of selenium and copper were observed in the shells of the gastropod *Rapana venosa*. The highest concentrations of chromium, manganese, and iodine were found in the shells of the ark clam *Anadara kagoshimensis*, and the highest concentrations of molybdenum were found in the shells of both *A. kagoshimensis* and the scallop *Flexopecten glaber ponticus*.

The principal component analysis demonstrated that the greatest contribution to the total variance in the concentrations was made by the overall accumulation of all the elements en masse due to random influences. The second most significant source of dispersion was the species specificity in the preferential accumulation of certain elements by specific mollusks.

The matrices of the pairwise Spearman’s correlations of the element concentrations showed the considerable degree of dissimilarity among the mollusks, which was confirmed by the cluster analysis. This result demonstrates the species-specific differences in the tendency for the concerted or competing accumulation of trace elements in the shells of these mollusks.

The present study allows for the assessment of the biotechnological potential of Black Sea mollusk shells as a source of essential elements. To meet the daily human requirement for essential trace elements, ground *Anadara*, *Crassostrea*, and *Rapana* shells have proven to be best suited. One needs to consume 2.7 and 25.6 g of ground *Anadara* shells to fulfill the daily requirements for iodine and chromium, respectively. *Rapana* shells can be used to supply the required daily dose of copper (8 g) and selenium (23.1 g). The amounts of 0.5 g and 18.3 g of *Crassostrea* shells are sufficient to compensate for the daily requirements for iron and cobalt, respectively.

## Figures and Tables

**Figure 1 animals-15-01637-f001:**
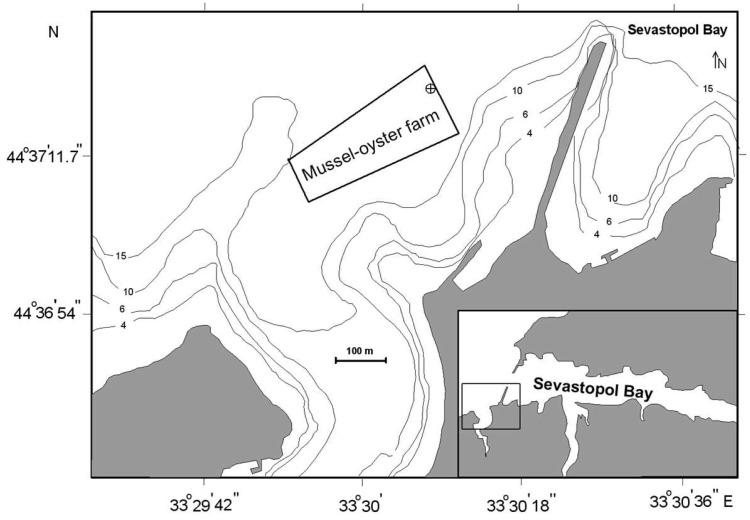
Map of sampling area. Sampling site is marked by circled cross.

**Figure 2 animals-15-01637-f002:**
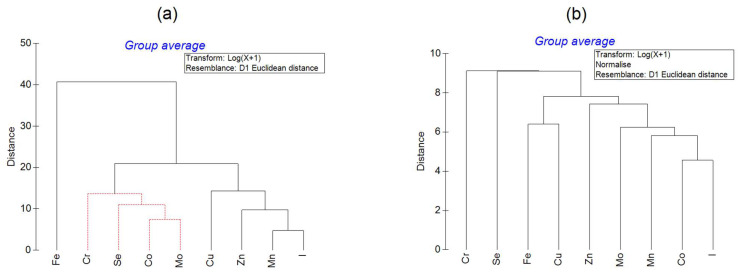
Dendrograms of the hierarchical cluster analysis of the log-transformed element concentrations (**a**) before and (**b**) after Z-standardization. Dashed lines indicate the lack of significance of cluster separation at the 5% level according to the SIMPROF test.

**Figure 3 animals-15-01637-f003:**
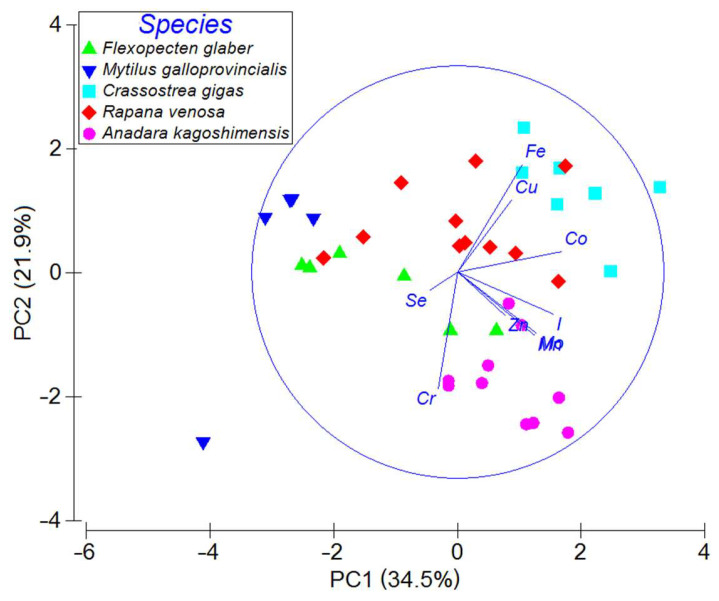
Principal component analysis of Z-standardized, log-transformed element concentrations in mollusk shells (legend).

**Figure 4 animals-15-01637-f004:**
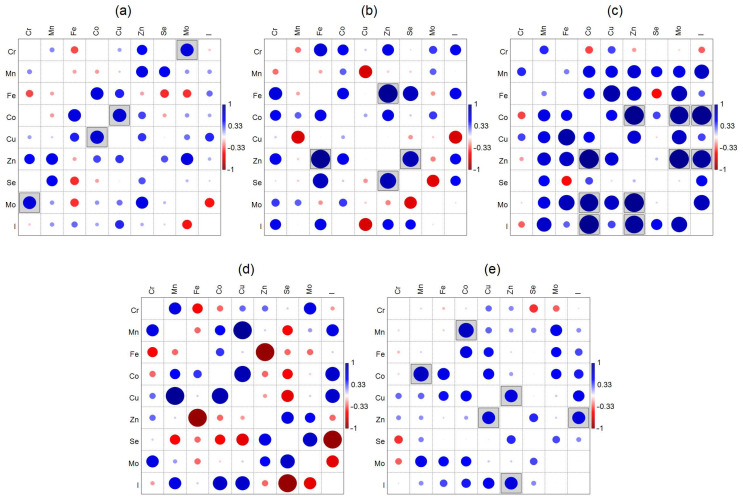
Graphical matrices of Spearman’s correlation coefficients (r_s_) for essential trace element concentrations in shells of (**a**) *Anadara kagoshimensis*, (**b**) *Crassostrea gigas*, (**c**) *Flexopecten glaber ponticus*, (**d**) *Mytilus galloprovincialis*, and (**e**) *Rapana venosa*. Significant correlations (*p* < 0.05) are marked by gray boxes.

**Figure 5 animals-15-01637-f005:**
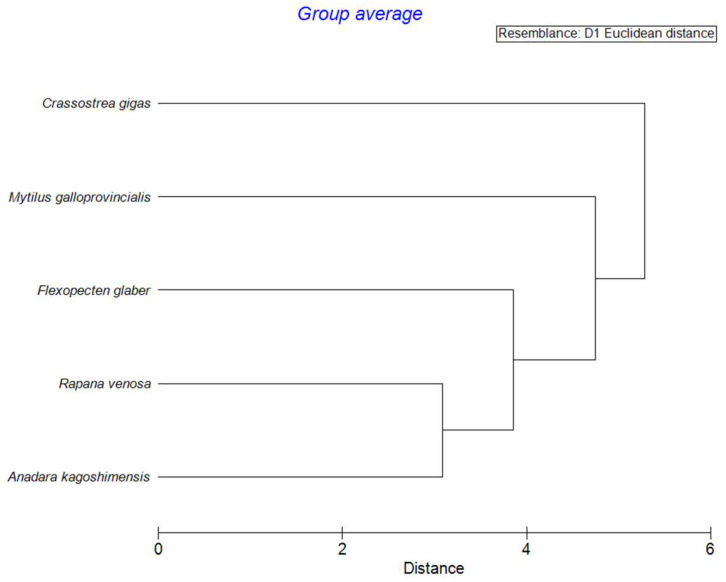
Cluster analysis of arrays (72 × 5) of Spearman’s coefficients of correlation between essential trace element concentrations in shells of five Black Sea mollusks.

**Table 1 animals-15-01637-t001:** Concentrations of essential trace elements (mg·kg^–1^ d.w.) in shells of Black Sea mollusks: mean ± standard deviation (min-max). * Significant differences from all mollusks, *^A^* significant differences from *Anadara*, *^F^* significant differences from *Flexopecten*, *^C^* significant differences from *Crassostrea*, *^M^* significant differences from *Mytilus*, *^R^* significant differences from *Rapana*. BDL = below detection level.

	*Anadara kagoshimensis*	*Crassostrea gigas*	*Flexopecten* *glaber ponticus*	*Mytilus galloprovincialis*	*Rapana venosa*
Cr	7.81 ± 4.60 * (2.80–15.2)	0.22 ± 0.28 *^AF^* (0.23–0.83)	1.74 ± 0.65 *^AC^* (1.30–2.83)	1.69 ± 1.45 *^A^* (0.98–4.30)	0.77 ± 0.50 *^A^* (BDL–1.70)
Mn	35.1 ± 18.2 (16.7–71.8)	24.3 ± 12.0 (14.9–49.5)	19.2 ± 5.3 (10.4–26.3)	8.17 ± 1.56 (6.20–10.3)	16.2 ± 10.9 (3.63–39.4)
Fe	1564 ± 325 ^C*RF*^ (117–2304)	13157 ± 4088 * (9391–21598)	2285 ±185 *^ACR^* (2023–2536)	1962 ± 1028 *^R^* (127–2490)	3294 ± 665 *^ACF^* (2335–4972)
Co	2.19 ± 0.14 *^F^* (1.91–2.44)	2.73 ± 0.97 (2.03–4.78)	1.29 ± 0.15 *^A^* (1.12–1.48)	0.99 ± 0.39 (0.29–1.19)	1.83 ± 0.63 (1.19–3.38)
Cu ^†^	75.0 ± 36.9 *^R^* (20.8–129)	170 ±73 (87.3–295)	89.4 ± 52.4 (46.2–178)	77.1 ± 21.0 (44.5–97.8)	190 ± 77*^A^* (22.2–303)
Zn	8.85 ± 8.26 (3.24–32.9)	21.1 ± 16.1 (4.71–63.3)	4.61 ± 3.27 (1.06–8.96)	30.0 ± 15.4 (1.26–69.1)	6.81 ± 2.64 (3.03–11.0)
Se	1.42 ± 0.95 (0.16–2.97)	1.31 ± 0.66 (0.26–10.5)	2.35 ± 0.26 (2.09–2.72)	2.43 ± 1.45 (1.05–4.92)	3.77 ± 3.41 (0.26–10.5)
Mo	1.00 ± 0.70 (0.50–2.92)	0.85 ± 0.41 (0.32–1.51)	1.06 ± 0.96 (0.24–3.06)	0.28 ± 0.12 (0.22–0.50)	0.56 ± 0.14 (0.26–0.87)
I	55.6 ± 17.3 *^FMR^* (39.0–82.1)	35.1 ± 5.4*^FM^* (30.6–45.1)	9.21 ± 4.83 *^AC^* (4.68–18.0)	5.90 ± 0.45 *^ACR^* (5.18–6.36)	26.0 ± 15.7 *^AM^* (10.7–67.4)

^†^ No significant heteroscedasticity.

## Data Availability

The data used in this study are available from the corresponding author on reasonable request.

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
