# Peer review of "Essential Trace Elements in the Shells of Commercial Mollusk Species from the Black Sea and Their Biotechnological Potential"

_animals, 2025, doi:10.3390/ani15111637_

Round 1

Reviewer 1 Report

Comments and Suggestions for Authors

This manuscript provides new data on the essential trace elements in shells of commercial mollusc species from the Black Sea and their biotechnological potential. The MS is generally well written with appropriate data analyses and an interesting discussion. Nevertheless, it has a number of shortcomings which should be addressed by the authors in order to improve understanding of the ms.

- There are analysis studies on Rapana venosa shells in the Black Sea. Why was this study needed?

- In addition to the lengths of the sample individuals, min-max lengths should also be given.

- Please explain whether, for each analytical sample, 5 replicates were selected from different individuals.

- It is seen that trace elements vary in different species. To meet daily human requirement, should trace elements (ground Anadara, Crassostrea and Rapana shells) be used separately or combined and how?

Author Response

Dear Reviewer #1

We are very grateful for your work aimed to help us improve the quality of our manuscript. The manuscript has been modified and supplemented with additional information according to Reviewers’ suggestions. All changes in the manuscript have been highlighted using the Track Changes function or, in rare cases (corrected citations), using red font. The changed citation numbering has not been highlighted to not overburden the manuscript with corrections. The responses to Reviewers’ comments and suggestions, point by point, are given below.

Kind regards,

Authors

Comments 1: “This manuscript provides new data on the essential trace elements in shells of commercial mollusc species from the Black Sea and their biotechnological potential. The MS is generally well written with appropriate data analyses and an interesting discussion. Nevertheless, it has a number of shortcomings which should be addressed by the authors in order to improve understanding of the ms.”

Response 1: We are thankful to the Reviewer for the positive reception of the manuscript and for the valuable suggestions.

Comments 2: “- There are analysis studies on Rapana venosa shells in the Black Sea. Why was this study needed?”

Response 2: Rapana venosa is one of the invasive dominant species in benthic ecosystems and a natural upper trophic link in the benthic communities of the Black Sea shelf, as there are no natural predators feeding on Rapana in the Black Sea ecosystem. In addition, it is a mollusc of industrial harvesting in the Black Sea. In 2016, the volume of Rapana production in the Black Sea amounted to 179.0 tons, and the mollusc fishery is currently active, requiring measures to be taken for utilizing shells of this gastropod. We are aware of as few as three studies on the elemental composition of shells of this mollusc, with the largest set of essential trace elements – Cr, Mn, Fe, Cu, Zn, and Mo – being studied in (Mititelu et al., 2008; Sereanu et al., 2018). References to these studies have been included in the revision. Unlike the previous studies by the Romanian researchers, this work is focused on quantifying the full suite of essential trace elements (in terms of the WHO classification) in shells of R. venosa.

Comments 3: “- In addition to the lengths of the sample individuals, min-max lengths should also be given.”

Response 3: The shell lengths of the selected bivalve individuals were as follows: 28–32 mm (30.5±1.0 mm) for Anadara kagoshimensis, 93–100 mm (90.7±10.3 mm ) for Crassostrea gigas, 31.1–45.5 mm (38.8±5.1 mm) for Flexopecten glaber ponticus, and 50–54.5 mm (54.1 ± 2.9 mm) for Mytilus galloprovincialis. The shell height of Rapana venosa was 77.5–89.4 mm (84.7±4.8 mm). The information on the min–max range has been added in the revision.

Comments 4: “- Please explain whether, for each analytical sample, 5 replicates were selected from different individuals.”

Response 4: As mentioned in Section 2.1, the elemental composition of each mollusc species was studied in 6–10 biological replicates (from the respective number of mollusc individuals). As regards the Reviewer’s comment, there were 5 replicates per analytical sample (Section 2.3) that were averaged to obtain the analytical sample mean, and there were two analytical samples prepared from each individual shell (Section 2.4) that were further averaged to yield the final concentration point used in the statistical treatment. This has been more explicitly stated in the respective sections. Thus, the numbers of biological samples, i.e. concentration points, were: 6 for the mussel and scallop, 7 for the oyster, and 10 for Anadara and Rapana.

Comments 5: “- It is seen that trace elements vary in different species. To meet daily human requirement, should trace elements (ground Anadara, Crassostrea and Rapana shells) be used separately or combined and how?”

Response 5: The results of the study indicate that the highest concentration of zinc is found in Mytilus shells, while selenium and copper are most concentrated in Rapana shells. Manganese, iodine, and chromium are predominantly present in Anadara shells, molybdenum is found in both Anadara and Flexopecten shells, and iron is concentrated in Crassostrea and Rapana shells. Obviously, preference should be given to molluscan shells with the highest levels of specific elements. It is not expected that different components of shells will exert considerable synergistic or antagonistic action on the element bioaccesability. Therefore, it is also possible to create compositions from different ground shells based on the linearity and proportionality of the element contents. From the daily human requirement for a certain element, it is easy to calculate how much pulverized shells should be consumed in the form of fortified food products (e.g. flour), food additives or encapsulated powdered functional products.

References

Mititelu, M.; Dogaru, E.; Nicolescu, T.O.; Hincu, L.; Băncescu, A.; Ioniţă, C. Heavy metals analysis in some mollusks shells from Black Sea. Scientific Study & Research 2008, 9(2), 195–198.

Sereanu, V.; Meghea, I.; Vasile, G.G.; Mihai, M. Environmental influence on Rapana venosa shell morphotypes and phenotypes from the Romanian Black Sea Coast. Revista de Chimie 2018, 69(1), 50–56. https://doi.org/10.37358/RC.18.1.6043

Reviewer 2 Report

Comments and Suggestions for Authors

Review for the paper submitted to “Animals”.

Title: Essential trace elements in shells of commercial mollusc species from the Black Sea and their biotechnological potential

Authors: Larisa L. Kapranova, Juliya D. Dikareva, Sergey V. Kapranov, Darya S. Balycheva, Vitaliy I. Ryabushko

The authors focused on commercial molluscs in the Black Sea, particularly highlighting species such as the ark clam Anadara kagoshimensis, the oyster Crassostrea gigas, the mussel Mytilus galloprovincialis, the scallop Flexopecten glaber ponticus, and the gastropod Rapana venosa. They conducted a comprehensive study analyzing the potential of mollusc shells for biotechnological applications, specifically in nutraceutical production. The authors' study showed that there are significant variations in the concentrations of essential trace elements in the shells of these five mollusc species collected from the same habitat.

There are the following implications of the authors' work: The study suggests that powdered shells from Anadara, Crassostrea, and Rapana could serve as effective dietary supplements for meeting human nutritional needs for essential trace elements. This positions these molluscs not only as commercial products but also as valuable resources for health and nutrition sectors. Moreover, the study underscores the potential for biotechnological applications of these shells, paving the way for innovative uses in various fields, including environmental sustainability and food technology.

The paper is well organized and coherently written. Only minor improvements are required

Suggestions for improving the paper:

Introduction.

L 64. The authors stated that mollusc shells decompose very slowly in the environment and impact ecosystems. They should explain what specific ecological impacts this slow decomposition can cause, such as changes to marine habitats or long-term accumulation of shell waste. This section should be supplemented with relevant citations.

L 68. The authors noted the nutritional applications of powdered mollusc shells for humans and animals. They should report on potential risks or limitations of introducing high concentrations of trace elements into the diet, especially regarding elements that could become toxic at excessive doses.

Materials and Methods.

L 101. Consider replacing “in the element distribution” with “on the element distribution”

L 105. The authors noted that the farm does not experience significant anthropogenic pressure throughout the year. They should explain what indicators or monitoring data support this claim.

L 108. The authors indicated that molluscs were collected at specific temperatures and depths in October 2022. They should clarify whether these environmental conditions are representative of the annual norms in the region.

L 113-116. What kind of error was used here (SD or SE)?

L 184-185. Statistical analysis included both Levene’s test and Welch’s ANOVA for heteroscedasticity scenarios. Can the authors specify which mollusc species comparisons required the Welch’s ANOVA and Games-Howell tests (i.e., where dispersion heterogeneities were found)?

L 193. Did the authors perform any corrections for multiple comparisons to reduce the likelihood of Type I errors, especially considering that several pairwise tests were conducted?

Results.

The authors use the present tense to describe the results. In scientific papers, the past tense is preferred.

Table 1. The authors should include the minimum and maximum values for element concentrations.

L 223. The text contains some details about cluster analysis that are not described in the Materials and Methods. The authors should describe the methodology in detail.

Discussion.

L 280-284. The citations referenced in the text are not included in the reference list.

L 286. The study implies that environmental and biological factors influence trace element deposition in shells. What environmental factors (e.g., salinity, temperature, or pollution levels) were correlated with shell composition in the study area?

L 308. What evidence supports the assumption about 100% bioaccessibility of trace elements from mollusk shells?

L 456. The text indicates that 283 g of mollusc shells is required to meet daily molybdenum needs, rendering shells impractical as a Mo source. Are there any documented methods in mineral extraction that could efficiently isolate molybdenum from shells for practical supplement production?

The authors should compare their concentrations with those reported for the same species from other regions and for other similar species.

Conclusions

This section should be shortened to focus on the main findings.

Author Response

Dear Reviewer #2,

We are very grateful for your work aimed to help us improve the quality of our manuscript. The manuscript has been modified and supplemented with additional information according to Reviewers’ suggestions. All changes in the manuscript have been highlighted using the Track Changes function or, in rare cases (corrected citations), using red font. The changed citation numbering has not been highlighted to not overburden the manuscript with corrections. The responses to Reviewers’ comments and suggestions, point by point, are given below.

Kind regards,

Authors

Response to the Comments of Reviewer #2

We warmly thank the Reviewer for the positive reception of the manuscript and valuable suggestions that have helped us to improve its contents and reliability.

Comments 1: “L 64. The authors stated that mollusc shells decompose very slowly in the environment and impact ecosystems. They should explain what specific ecological impacts this slow decomposition can cause, such as changes to marine habitats or long-term accumulation of shell waste. This section should be supplemented with relevant citations.”

Response 1: When processing shellfish, the mass of shells makes up about 60–80% of the total mass (Suresh, Prabhu, 2013). Therefore, a significant portion of the shellfish farm output is considered nuisance waste. Without regulated disposal procedures, shell waste is frequently discarded in landfills or dumped at sea. Shell waste accumulates and pollutes coastal areas, emitting an unpleasant odor due to the decomposition of organic matter trapped within the shells (Hou et al., 2016; Morris et al., 2018; Jović et al., 2019).

The effects of increased shell deposition beneath cages and shell accumulation on shellfish farms are not well understood, but these environmental impacts can be both positive and negative (Sanchez-Jerez et al., 2019). On the one hand, the accumulated shells are a crucial habitat for macrozoobenthos. Mollusc shells serve as a substrate for the attachment of epibionts and shelter from predators, and they affect the transport of materials in the benthic environment (Gutiérrez et al., 2003). On the other hand, fragmented shells reduce animal diversity and abundance, e.g. by preventing oxygen from penetrating into the underlying sediments. The effects of suspended oyster farming on accumulation of shells on the seafloor and their impact on the burrowing ability of the polychaete Perinereis aibuhitensis were studied. The presence of unprocessed oyster shells inhibited the burrowing activity of P. aibuhitensis (Cao et al., 2023).

The above information has been added to Introduction.

Comments 2: “L 68. The authors noted the nutritional applications of powdered mollusc shells for humans and animals. They should report on potential risks or limitations of introducing high concentrations of trace elements into the diet, especially regarding elements that could become toxic at excessive doses.”

Response 2: Processed mollusc shells can be used as additives in animal feed (Khalil et al., 2018) or as food supplements for humans. The need for essential trace elements depends on the age and sex of the person, the species of the animal, and its level of productivity. In highly productive and intensively growing animals, as well as in young animals, it is significantly higher. Elements such as iron, copper, zinc, cobalt, manganese, iodine and selenium are important for feeding poultry. Cattle are supplied with standardized amounts of iron, copper, zinc, cobalt, manganese and iodine in their diet. It is important to take into account that marine organisms are capable of bioconcentrating heavy metals along the food chain (Ozuni et al., 2024). Therefore, consuming seafood and processed products containing heavy metal concentrations that exceed permissible levels may pose a health risk. This also applies to essential elements: even these elements can be toxic if taken in large amounts (Tüzen, 2003; Beldi, et al., 2006; Keskin et al., 2015). To minimize the risks associated with shellfish consumption, it is advisable to monitor the levels of metals in shellfish products. This monitoring allows for comparison with established safe limits for the concentration and dosage of elements in food consumed. For example, the safe concentration for Cd is (1.01–2.00 μg/g) (CE, 2006), while the concentration 5.01–10.00 μg/g is considered hazardous (Hossen et al., 2015). The minimum safety limit for zinc is 100.01–250.00 μg/g, and the highest safety limit is 667.01–750.00 μg/g (Hossen et al., 2015).

We added some of this information to Introduction as suggested.

Comments 3: “L 101. Consider replacing “in the element distribution” with “on the element distribution””

Response 3: “Influence on the element distribution” suggested by the Reviewer is not what was meant here. “Non-uniformities in the element distribution” is the accurate phrase in this sentence. The influence of local non-uniformities on the element concentration in the molluscs is tacitly implied.

Comments 4: “L 105. The authors noted that the farm does not experience significant anthropogenic pressure throughout the year. They should explain what indicators or monitoring data support this claim.”

Response 4: There are no industrial enterprises near the farm. The source of pollution may be currents from the Sevastopol Bay that carry pollutants outside it (Kapranov et al., 2020). In the marine farm area, sedimentation processes make a substantial contribution to the purification of waters by removing a substantial quantity of toxic elements (Pospelova et al., 2022). This information has been added to the manuscript.

Comments 5: “L 108. The authors indicated that molluscs were collected at specific temperatures and depths in October 2022. They should clarify whether these environmental conditions are representative of the annual norms in the region.”

Response 5: The temperature and salinity values recorded during the sampling were perceptibly higher than the mean ones recorded in October in the period of 2001-2018, 17.5 °C and 17.8 psu (Kapranov et al., 2020), and they were even above the maximum limits for that period (18 °C and 17.96 psu). The increased temperature and salinity values were due to the steady increasing trends of these parameters over the past few decades in this area (Kapranov et al., 2020). This information has been added in the revision.

Comments 6: “L 113-116. What kind of error was used here (SD or SE)?”

Response 6: It was SD. This has been mentioned in the revision.

Comments 7: “L 184-185. Statistical analysis included both Levene’s test and Welch’s ANOVA for heteroscedasticity scenarios. Can the authors specify which mollusc species comparisons required the Welch’s ANOVA and Games-Howell tests (i.e., where dispersion heterogeneities were found)?”

Response 7: All elements except Cu yielded significantly heterogeneous dispersions, according to Levene’s test. Copper has been marked in the revised Table 1 as the element with the lack of heterogeneous dispersions. Unfortunately, Levene’s test does not provide information on which mollusc(s) are responsible for the heteroscedasticity. Nor were any leave-one-out trial comparisons conducted. Visual comparison suggests that the highest variances apparently significantly exceeding the others were attributed to the following molluscs: Cr, Mn, and I – Anadara; Fe –Crassostrea; Co – Crassostrea and Rapana; Zn – Mytilus; Se – Rapana; Mo – Flexopecten, Anadara, and Crassostrea.

Comments 8: “L 193. Did the authors perform any corrections for multiple comparisons to reduce the likelihood of Type I errors, especially considering that several pairwise tests were conducted?”

Response 8: We thank the Reviewer for pointing this out. In the revision, corrections for multiple comparisons have been made using the Benjamini-Hochberg method, as mentioned in Section 2.5. It proves to be more powerful (less conservative against Type I errors) than the Holm-Bonferroni method.

Comments 9: “The authors use the present tense to describe the results. In scientific papers, the past tense is preferred.”

Response 9: We agree with the Reviewer in that the past tense is more appropriate for describing data that have been obtained “behind the scene”, when the reader sees only the final result. For example, this is relevant to describing the ICP-MS analysis results. However, we do not consider it appropriate to use the past tense when the reader can see the entire process in which the results have been obtained or the generality of the result. For instance, it seems awkward to write: “Hierarchical cluster analysis… showed that the elements formed three distinct clusters,” because the reader sees the cluster analysis dendrogram and how the clusters are partitioned in it. Likewise, in the statement: “It is thus evident that oysters are very effective at accumulating most of the metallic elements and iodine,” the past tense does not seem apposite, as the oysters are, in general, effective element bioaccumulators. Therefore, the past tense has been sparingly used in the revision in accordance to the above-mentioned principles.

Comments 10: “Table 1. The authors should include the minimum and maximum values for element concentrations.”

Response 10: The minimum and maximum values for the element concentrations have been included in Table 1.

Comments 11: “L 223. The text contains some details about cluster analysis that are not described in the Materials and Methods. The authors should describe the methodology in detail.”

Response 11: The Z-standardization applied to the concentration data prior to running PCA and cluster analysis in Figure 2b has been explained in the Materials and Methods section.

Comments 12: “L 280-284. The citations referenced in the text are not included in the reference list.”

Response 12: The respective citations and references have been corrected (p. 9).

Comments 13: “L 286. The study implies that environmental and biological factors influence trace element deposition in shells. What environmental factors (e.g., salinity, temperature, or pollution levels) were correlated with shell composition in the study area?”

Response 13: Our study did not examine element accumulation in shells as a function of environmental parameters, as all the bivalves had been in the same environmental conditions (and Rapana in approximately the same conditions as for the bivalves).

It is known that the chemical composition of the shell is indeed affected by many external factors, such as water temperature, salinity, pH, dissolved oxygen concentration, currents, coastal pollution etc. These influences are imprinted in the growth rate, calcium carbonate biodeposition, and the incorporation of trace elements into the shell structure (Forleo et al., 2021). Increased accumulation of heavy metals was noted in contaminated areas (Vieira et al., 2021). Temperature also affects the solubility of heavy metals, increasing their bioavailability to molluscs.

Comments 14: “L 308. What evidence supports the assumption about 100% bioaccessibility of trace elements from mollusk shells?”

Response 14: The assumption of 100% element bioaccessibility may, of course, be rather far from realistic values; however, this is merely a rough estimate of biotechnological applicability of shells as essential trace element nutraceuticals, as emphasized in the revision. The degree of assimilation of elements after the ingestion may vary widely (Amiard et al., 2008). For example, for lead it ranges from 3% to 80%. It is greatly affected by food intake, with the degree of lead absorption after fasting being significantly higher than during the normal food intake. The absorption is also influenced by age, with typical absorption rates in adults and infants being 10% and 50%, respectively (WHO, 2000).

In individuals on a low copper diet (0.78 mg/day), the Cu assimilation was 55.6%, compared to only 36.3% at the recommended total daily intake (1.68 mg/day) and 12.4% for the high copper diet (7.53 mg/day). People with adequate dietary zinc assimilate approximately 20–30% of the total zinc consumed, while people with dietary zinc deficiency assimilate more zinc if it is in a bioavailable form (WHO, 2001).

Addition of shell powder to the diet of hens significantly increased their morphometric parameters and egg production, likely indicating high bioavailability of elements in this matrix (Aletor, Onibi, 1990; Cho et al., 2017).

Comments 15: “L 456. The text indicates that 283 g of mollusc shells is required to meet daily molybdenum needs, rendering shells impractical as a Mo source. Are there any documented methods in mineral extraction that could efficiently isolate molybdenum from shells for practical supplement production?”

Response 15: At present, extraction of molybdenum from mollusc shells is not studied, and no methods for concentrating Mo in food supplements from mollusc shells have been found in the literature. The elaboration of such technologies requires novel methodological designs, including ones from the perspective of sustainable development and marketing efficiency.

Comments 16: “The authors should compare their concentrations with those reported for the same species from other regions and for other similar species.”

Response 16: According to the Reviewer’s suggestion, a new subsection 4.2 has been introduced to compare our results with the literature data on essential trace elements in shells of the same or similar species from other regions, while the rest of Discussion migrated into the newly established subsections 4.1 and 4.3.

Comments 17: “Conclusions This section should be shortened to focus on the main findings.”

Response 18: This section has been shortened by removing the last paragraph that was of little relevance to the results of this study.

REFERENCES

Aletor, V.A.; Onibi, O.E. Use of oyster shell as calcium supplement. Part 1. Effect on the utilization of gossypol containing cotton seed cake by the chicken. Food/Nahrung 1990, 34(4), 311–318. https://doi.org/10.1002/food.19900340403

Amiard, J.C.; Amiard-Triquet, C.; Charbonnier, L.; Mesnil, A.; Rainbow, P.S.; Wang, W.X. Bioaccessibility of essential and non-essential metals in commercial shellfish from Western Europe and Asia. Food and Chemical Toxicology 2008, 46(6), 2010-2022. https://doi.org/10.1016/j.fct.2008.01.041

Beldi, H.; Gimbert, F.; Maas, S.; Scheifler, R.; Soltani, N. Seasonal variations of Cd, Cu, Pb and Zn in the edible mollusc Donax trunculus (Mollusca, Bivalvia) from the gulf of Annaba, Algeria. African Journal of Agricultural Research 2006, 1(4), 85–90.

Cao, Y.; Shi, R.; Han, T.; Liu, H.; Huang, H.; Qi, Z. Shell accumulation on seabed due to suspended coastal oyster farming and effects on burrowing capacity of the polychaete Perinereis aibuhitensis. Frontiers in Marine Science 2023, 10, 1219184. https://doi.org/10.3389/fmars.2023.1219184

Cho, M.G.; Bae, S.M.; Jeong, J.Y. Egg shell and oyster shell powder as alternatives for synthetic phosphate: Effects on the quality of cooked ground pork products. Korean Journal for Food Science of Animal Resources 2017, 37(4), 571–578. https://doi.org/10.5851/kosfa.2017.37.4.571

Forleo, T.; Zappi, A.; Melucci, D.; Ciriaci, M.; Griffoni, F.; Bacchiocchi, S.; Siracusa, M.; Tavoloni, T.; Piersanti, A. Inorganic elements in Mytilus galloprovincialis shells: Geographic traceability by multivariate analysis of ICP-MS data. Molecules 2021, 26, 2634. https://doi.org/10.3390/molecules26092634

Gutiérrez, J.L.; Jones, C.G.; Strayer, D.L.; Iribarne, O.O. Mollusks as ecosystem engineers: the role of shell production in aquatic habitats. Oikos 2003, 101, 79–90. https://doi.org/10.1034/j.1600-0706.2003.12322.x

Hossen, M.F.; Hamdan, S.; Rahman, M.R. Review on the risk assessment of heavy metals in Malaysian clams. Scientific World Journal 2015, 2015, 905497. https://doi.org/10.1155/2015/905497

Hou, Y.; Shavandi, A.; Carne, A.; Bekhit, A.A.; Ng, T.B.; Cheung, R.C.F.; Bekhit, A.E.D.A. Marine shells: Potential opportunities for extraction of functional and health-promoting materials. Critical Reviews in Environmental Science and Technology 2016, 46, 1047–1116. https://doi.org/10.1080/10643389.2016.1202669

Jović, M.; Mandić, M.; Šljivić-Ivanović, M.; Smičiklas, I. Recent trends in aplication of shell waste from mariculture. Studia Marina 2019, 32(1), 47–62. https://doi.org/10.5281/zenodo.3274471

Kapranov, S.V.; Kovrigina, N.P.; Troshchenko, O.A.; Rodionova, N.Y. Long-term variations of thermohaline and hydrochemical characteristics in the mussel farm area in the coastal waters off Sevastopol (Black Sea) in 2001–2018. Continental Shelf Research 2020, 206, 104185. https://doi.org/10.1016/j.csr.2020.104185

Keskin, Y.; Baskaya, R.; Özyaral, O.; Yurdun, T.; Lüleci, N.E.; Hayran, O. Cadmium, lead, mercury and copper in fish from the Marmara Sea, Turkey. Bulletin of Environmental Contamination and Toxicology 2007, 78, 258–261. https://doi.org/10.1007/s00128-007-9123-9

Khalil; Wati, W.; Hidayat, F.; Evitayani. Physical Properties and Nutritive Values of Shell Meal Derived from Different Shellfish Species and Habitats. International Journal of Poultry Science 2018, 17(3), 116–125. https://doi.org/10.3923/ijps.2018.116.125

Morris, J.P.; Backeljau, T.; Chapelle, G. Shells from aquaculture: A valuable biomaterial, not a nuisance waste product. Reviews in Aquaculture 2018, 11, 42–57. https://doi.org/10.1111/raq.12225

Ozuni, E.; Andoni, E.; Castrica, M.; Balzaretti, CM.; Brecchia, G.; Agradi, S.; Curone, G.; Di Cesare, F.; Fehri, N.E.; Luke, B.; Erman Or, M.; Akkaya, E.; Yavuz, O.; Menchetti, L.; Prendi, L.; Özsonacı, N.P.; Ercan, A.M.; Ateş, F.; Miraglia, D. Human exposure to heavy metals and possible public health risks via consumption of mussels M. galloprovincialis from the Albanian sea coast. Chemosphere 2024, 368, 143689. https://doi.org/10.1016/j.chemosphere.2024.143689

Pospelova, N.V.; Egorov, V.N.; Proskurnin, V.Y.; Priimak, A.S. Suspended particulate matter as a biochemical barrier to heavy metals in marine farm areas (Sevastopol, the Black Sea). Marine Biological Journal 2022, 7(4), 55–69. https://doi.org/10.21072/mbj.2022.07.4.05

Sanchez-Jerez, P.; Krüger, L.; Casado-Coy, N.; Valle, C.; Sanz-Lazaro, C. Mollusk shell debris accumulation in the seabed derived from coastal fish farming. Journal of Marine Science and Engineering 2019, 7, 335. https://doi.org/10.3390/jmse7100335

Suresh, P.V.; Prabhu, G.N. Seafood. In Valorization of Food Processing By-Products; Chandrasekaran, M., Ed.; CRC Press: Boca Raton, FL, 2013; pp. 685–736.

Tüzen, M. Determination of heavy metals in fish samples of the middle Black Sea (Turkey) by graphite furnace atomic absorption spectrometry. Food Chemistry 2003, 80, 119–123. https://doi.org/10.1016/S0308-8146(02)00264-9

Vieira, K.S.; Crapez, M.A.C.; Lima, L.S.; Delgado, J.F.; Brito, E.B.C.C. ; Fonseca, E.M.; Baptista Neto, J.A. ; Aguiar, V.M.C.. Evaluation of bioavailability of trace metals through bioindicators in a urbanized estuarine system in southeast Brazil. Environmental Monitoring and Assessment 2021, 193, 18. https://doi.org/10.1007/s10661-020-08809-x

WHO. Safety evaluation of certain food additives and contaminants. 53rd Report of the Joint FAO/WHO Expert Committee on Food Additives, Geneva. WHO Food Additives Series No. 44, 2000.

WHO. Zinc. Report of IPCS (Joint UNEP/ILO/WHO), Geneva. WHO, Environmental Health Criteria, No. 221, 2001.

Round 2

Reviewer 1 Report

Comments and Suggestions for Authors Necessary edits have been made to the text.
Therefore, the study can be accepted as a research article.